# The effects of cholesterol accumulation on Achilles tendon biomechanics: A cross-sectional study

Kipling Squier[1,2]*, Alexander Scott[1,2], Michael A. Hunt[1,2], Liam R. Brunham[3], David R. Wilson[2,4], Hazel Screen[5], Charlie M. Waugh[1,2]

**1** Department of Physical Therapy, Faculty of Medicine, University of British Columbia, Vancouver, British Columbia, Canada, **2** Centre for Hip Health and Mobility, Vancouver Coastal Health Research Institute, Vancouver, British Columbia, Canada, **3** Centre for Heart Lung Innovation, University of British Columbia, Vancouver, British Columbia, Canada, **4** Department of Orthopaedics, Faculty of Medicine, University of British Columbia, Vancouver, British Columbia, Canada, **5** School of Engineering & Materials Science, Queen Mary University of London, London, United Kingdom

* kip.squier@hiphealth.ca

**Data Availability Statement:** The data used in this study are available on the OSF repository at: https://osf.io/8tq7v/?view_only=

## Abstract

Familial hypercholesterolemia, a common genetic metabolic disorder characterized by high cholesterol levels, is involved in the development of atherosclerosis and other preventable diseases. Familial hypercholesterolemia can also cause tendinous abnormalities, such as thickening and xanthoma (tendon lipid accumulation) in the Achilles, which may impede tendon biomechanics. The objective of this study was to investigate the effect of cholesterol accumulation on the biomechanical performance of Achilles tendons, *in vivo*. 16 participants (10 men, 6 women; 37±6 years) with familial hypercholesterolemia, diagnosed with tendon xanthoma, and 16 controls (10 men, 6 women; 36±7 years) underwent Achilles biomechanical assessment. Achilles biomechanical data was obtained during preferred pace, shod, walking by analysis of lower limb kinematics and kinetics utilizing 3D motion capture and an instrumented treadmill. Gastrocnemius medialis muscle-tendon junction displacement was imaged using ultrasonography. Achilles stiffness, hysteresis, strain and force were calculated from displacement-force data acquired during loading cycles, and tested for statistical differences using one-way ANOVA. Statistical parametric mapping was used to examine group differences in temporal data. Participants with familial hypercholesterolemia displayed lower Achilles stiffness compared to the control group (familial hypercholesterolemia group: 87±20 N/mm; controls: 111±18 N/mm; $p = 0.001$), which appeared to be linked to Achilles loading rate rather than an increased strain (FH: 5.27±1.2%; controls: 4.95±0.9%; $p = 0.413$). We found different Achilles loading patterns in the familial hypercholesterolemia group, which were traced to differences in the centre of pressure progression that affected ankle moment. This finding may indicate that individuals with familial hypercholesterolemia use different Achilles loading strategies. Participants with familial hypercholesterolemia also demonstrated significantly greater Achilles hysteresis than the control group (familial hypercholesterolemia: 57.5±7.3%; controls: 43.8±10%; $p<0.001$), suggesting that walking may require a greater metabolic cost. Our results indicate that cholesterol

3b96175e561b4366bd7a51e62b0f2709 (DOI: 10. 17605/OSF.IO/8TQ7V).

**Funding:** CW received funding from the European Union's Horizon 2020 research and innovation programme under the Marie Sklodowska-Curie grant agreement (No. 704333). AS received funding from the Canadian Institutes of Health Research (PJT-166129). The funders had no role in study design, data collection and analysis, decision to publish, or preparation of the manuscript.

**Competing interests:** The authors have declared that no competing interests exist.

accumulation could contribute to reduced Achilles function, while potentially increasing the chance of injury.

## Introduction

Affecting 1 in 250 people, heterozygous familial hypercholesterolemia (FH) is the most common genetic disorder concerning metabolism of low density lipoproteins cholesterol (LDL) [1]. Due to mutations in genes affecting the LDL receptor, hepatic uptake of LDL particles is impaired, leading to elevated serum LDL cholesterol levels [2]. While cholesterol is essential for the normal functioning of cell membranes and hormone production, elevated LDL cholesterol in plasma leads to deposition of cholesterol-rich plaque in arterial walls, resulting in coronary heart disease (CHD) [3, 4]. In addition to its role in atherosclerosis, several studies have shown that high levels of LDL cholesterol could increase rates of tendinopathy, which is likely due to manifestation and accumulation of LDL within a tendon's extracellular matrix (ECM) [5–7].

In people with FH, tendinous cholesterol accumulations can be so severe that they develop diffuse thickening, as well as areas of concentrated thickening, known as tendinous xanthoma (TXT) [8]. TXT most commonly occur in the Achilles tendon (AT), resulting in persistent low-grade inflammation and pain [8–10]. Formation of TXT within the tendon could be due to a strong affinity between a tendon's increased collagen content and LDL lipids, resulting in LDL leaving the bloodstream to enter the tendon ECM [11].

High-strain tendons may present with TXT more frequently than positional tendons, which could stem from their high glycosaminoglycan (GAG) content compared to positional tendons [12, 13]. Tendon extension occurs through substantial sliding of different hierarchical collagen units against each other, which is facilitated by the GAG-rich non-collagenous matrix [14]. As LDL has a high affinity for GAGs, the extension mechanism of high-strain tendons that have a high GAG content (such as the Achilles tendon) may well be impacted in a high cholesterol environment [15].

TXT could have a profound negative consequence on the performance of human energy storing tendons, particularly the AT, which relies heavily on ECM composition to maintain sufficient stiffness and elastic properties to withstand large forces [16, 17]. The AT is unique due to its ability to store and return plantar flexion energy which minimizes the muscle's metabolic cost during ambulation [18–20]. During level walking, the triceps surae muscles contract near isometrically, and the majority of the muscle-tendon unit (MTU) displacement occurs through AT extension [21]. The AT is gradually loaded through the bulk of the stance phase to store energy. At toe off the tendon rapidly recoils to return stored energy [19]. However, the AT's ability to store and return energy is not perfect; the tendon dissipates energy during unloading in the form of hysteresis. Changes in the composition of the AT ECM could alter the biomechanical properties of the tendon and impact energy storage and recovery [22, 23].

Research into the effects of cholesterol accumulation on tendon biomechanics has been primarily conducted on *in vitro* animal models, producing mixed results. Beason *et al.* reported that supraspinatus tendons were inclined to have an increased stiffness and modulus in animals presenting with lipid accumulation: genetically modified mice (apolipoprotein E knockout); monkeys (high cholesterol diet) and rats (high cholesterol diet) [24]. Contrary to these findings, the same researchers found that there was a significant drop in stiffness and modulus in high cholesterol porcine bicep tendons, and mouse patellar tendons [25, 26]. Similarly, Grewal *et al.* found that patellar tendons of high fat dieted mice failed at significantly lower loads

and stresses [13]. Differences between study's findings could be associated with mechanical testing methodology, or result from differences in species, tendon type and methodology for cholesterol exposure [25].

Given that TXT are commonly found in human AT, we propose that this would be a suitable tendon to investigate the impact of cholesterol accumulation on tendon biomechanics in humans [8]. Appropriate AT function is vital for everyday ambulation, thus the current study aims to explore whether cholesterol accumulation modifies AT biomechanical properties and behaviour during walking. We hypothesized that persons with FH will demonstrate altered AT mechanical properties and function when compared to a control group (CG) and specifically sought to examine AT stiffness, strain, and hysteresis during the cyclic loading of walking.

## Methods

### Participants

16 participants with FH (10 men and 6 women, age: 37±6.1, BMI: 28.2±4 kg/m2 [83.5±13.85 kg]) were recruited from The BC FH Registry (Centre for Heart Lung Innovation, St Paul's Hospital, Vancouver, Canada) and 16 healthy, gender- and BMI-matched CG participants (10 men and 6 women, age: 36±6.6, BMI: 26.7±2.5 kg/m2 [80.9±15 kg]) were recruited using convenience sampling. For inclusion, participants with FH must have been diagnosed with "definite" FH, determined by the Dutch Lipid Clinic Network Score (a score of ≥8 points); and have been diagnosed with Achilles TXT in a physical examination performed by a physician at the Centre for Heart Lung Innovation. All participants were required to be between 19–50 years old at the time of testing, be able to participate in moderate physical activity, and have a BMI of under 35. The participants were excluded if they had prior acute injuries to the AT, chronic conditions that could negatively influence musculoskeletal properties, recent lower limb musculoskeletal injuries, or participated in a structured resistance training specifically targeting the triceps surae in the year preceding data collection. All participants provided written informed consent prior to participation in the current study. The study was approved by the Clinical Research Ethics Board, University of British Columbia (#H16-01358). Study testing took place at the Centre for Hip Health and Mobility (Vancouver, Canada).

### Experimental protocol

Participants were monitored through preferred pace, shod, walking trials using modified inverse dynamics methodology, similar to work performed by Lichtwark & Wilson [27]. Participants walked in their favored athletic footwear on a fully instrumented treadmill (Bertec, Columbus, OH, USA) for a 10-minute acclimatization period, to familiarize themselves with treadmill gait style, split belt hardware and precondition the AT to account for viscoelastic behaviors [28]. To keep participants centered on the treadmill and to divert attention away from their gait, a centered vertical line was positioned 3 meters in front of the treadmill for them to focus on. Preferred pace was determined by finding a pace that describes the participant's everyday walking velocity while "walking to work" or "doing chores". To assist the participant in identifying the velocity, the treadmill belt speed started slow and was incrementally increased until the participant professed a fast walk was achieved. Belt speed was then lowered until the preferred pace was attained. At each change of belt speed, the participant was asked about their comfort and how the speed related to their normal pace. After the acclimatization period, 15 seconds of synchronized data (kinematics, kinetics and ultrasound) were collected on the participant's self defined dominant limb.

**Clinical characteristics.** Participants with FH had their clinical characteristics (most recent serum analysis) retrieved from the BC FH Registry. Lipid lowering medication usage

and serum levels of LDL, total cholesterol, and apolipoprotein B were noted. Clinical characteristics for CG are unfortunately not available for comparison.

**Kinematics.**   Kinematic data were collected by tracking passive markers with eight ceiling-mounted infrared cameras (4 Raptor-E & 4 Raptor-H, Motion Analysis, Santa Rosa, CA, USA). Motion capture markers were fixed to the participant's dominant leg using Transpore™ tape (3M, St. Paul, MN, USA). Skin-based motion capture markers were fixed to the first metatarsal head, calcaneus medial and lateral malleoli, head of the fibula, and lateral epicondyle of the femur. A four-marker cluster was also permanently fixed to the ultrasound probe housing. The foot was assumed to be a rigid segment between the heel and toe markers. Marker coordinates were sampled at 150 Hz and were smoothed using a low-pass, zero-lag Butterworth filter at 8 Hz. Ankle angle was defined as the acute angle between the lateral femoral condyle-lateral malleolus-fifth metatarsal head markers. A virtual marker was positioned at the superior edge of the ultrasound scanning interface to simulate the origin of the ultrasound reference frame, whose position was calculated relative to the rigid body cluster attached to the ultrasound probe casing. The virtual marker allowed the GM MTJ to be assigned 3D coordinates within the motion capture reference frame.

**Kinetics.**   The instrumented treadmill collected the gait ground reaction forces (GRF) at 1500Hz and was calibrated within the motion capture space. Kinetic data was synchronized with kinematic and ultrasound data through a 16-bit A/D card (NI USB-6218, National Instruments, Austin, TX, USA), using a custom-built analog synchronizing trigger. System and trigger delays were incorporated into the data synchronisation process.

**Ultrasonography.**   B-mode ultrasound with a 60 mm linear array probe (RP Sonix, Ultrasonix, Burnaby, BC, Canada) was used to measure two-dimensional displacement of the gastrocnemius medialis (GM) muscle-tendon junction (MTJ). The ultrasound was set to 10 MHz scanning frequency, 60 Hz sampling frequency and 3.0 cm scanning depth. The ultrasound probe was tightly housed in a custom 3D printed casing, which was securely fixed over the GM MTJ, just medial to the septum with the gastrocnemius lateralis, on the participant's dominant leg. Displacement of the GM MTJ was tracked manually with open-source software (Tracker v4.9.8, physlets.org/tracker). Data were low-pass filtered using a fourth-order, zero-lag Butterworth filter with a 3.5-Hz cut-off frequency, as determined by residual analysis.

## Variable calculation

All variables were calculated using a custom-written analysis program (Matlab v14, Mathworks, Cambridge, United Kingdom). The first eight consecutive stance phases of each participant's walking trial were analyzed. Stance phases that showed GRF contamination from the contralateral limb were rejected and replaced by subsequent uncontaminated stance phases. Heel strike and toe off were defined as the instant where the vertical GRF exceeds or returns to within two standard deviations above baseline, respectively.

AT deformation was calculated using kinematics and ultrasound imaging. AT length was calculated as the linear distance from the calcaneus to the instantaneous position of the 3D GM MTJ coordinates. AT force was calculated as the ratio of instantaneous plantarflexion moment to instantaneous AT moment arm length. Plantarflexion moment was calculated in the sagittal plane using inverse dynamics principles. The ankle joint centre of rotation was represented by a virtual marker positioned equidistant between the medial and lateral malleoli markers. Instantaneous AT moment arm length was calculated as the shortest perpendicular distance from the ankle joint centre of rotation to the AT vector line of action (calcaneus to 3D GM MTJ). Because the calcaneus marker was placed on exterior of shoe, the distance between skin and marker was measured with a caliper and incorporated into the AT moment arm

length calculation. To avoid underestimating the external moment arm by the toe-out or toe-in positioning of the foot during the stance phase, the mean angle difference of the long foot axis in relation to the sagittal plane was determined and the external moment arm adjusted accordingly.

Stiffness (N/mm) was calculated 1) continuously over the ascending slope of the AT force-deformation curve using a 5-data point moving average ($k_{cont}$), and 2) in the region between mid and terminal stance in which the linear force-extension behaviour is seen, between 50–100% of peak AT force ($k_{50-100\%}$). Peak strain (% change in length) was calculated as peak AT length in relation to AT length at heel strike. AT hysteresis (% of total energy lost) was defined as the difference between the area under the ascending and descending portions of the force-deformation curve. Typically, Young's modulus would be used as an appropriate measure for comparing tendon properties by removing the influence of tendon dimensions. However, we chose not to use measures that incorporate CSA measures, as TXT may not hold any load bearing qualities and would therefore skew the results and comparison between the two groups [29]. All variables were calculated for the ground contact portion of the gait cycle only.

## Statistical analysis

Variable data were analyzed using SPSS statistical software (SPSS v.20, IBM Corp., Armonk, NY, USA). To test for significant differences in stiffness, strain and hysteresis between groups, a one-way ANOVA was performed. In case of significance, these dependent variables were examined using independent t-tests. Independent t-tests using statistical parametric mapping (SPM) (spm1d v.0.4, www.spm1d.org) were performed on temporal data. AT force and strain, ankle angle and $k_{cont}$ curves were examined this way to identify whether statistical differences occurred across the stance phase [30]. $k_{cont}$ was only examined with SPM between 20–70% stance due to extreme data variability with AT force/displacement curve inflection points outside this range. MANOVA analysis was performed to determine if sex played a role in the effect of cholesterol accumulation on the primary outcomes. Significance was established to be $p \leq 0.05$ for all statistical tests.

## Results

32 participants (16 participants with FH, 16 CG participants) participated in the study. Each participant with FH was noted to be positive for TXT within the registry medical charts and all but one participant with FH were currently using one or more lipid lowering medications (mean usage 7.7±7.2 years). Clinical characteristics for participant's with FH (CG not available) retrieval revealed a mean Dutch Lipid Clinic Network Score of 18±6 and a moderately controlled lipid profile (LDL: 3.5±1.6 mmol/L, total cholesterol: 5.4±1.6 mmol/L, apolipoprotein B:1.02±0.36 g/L). Of the 32 participants, 2 participants (1 FH and 1 CG) were left leg dominant. Average AT length at heel strike was similar between FH (218±28 mm) and CG (211±34 mm). There was not a significant difference in preferred walking pace between FH (1.17±0.1 m/s) and CG (1.22±0.1 m/s).

Visual inspection of the load (normalized to peak AT force)—strain curves between the two groups (Fig 1) highlights the reduced incline of the FH ascending slope, indicating decreased stiffness particularly between 50–100% AT force; CG demonstrated less area between the ascending and descending portions of the curve, indicating a smaller hysteresis than FH.

Group mean values for AT stiffness ($k_{50-100\%}$), hysteresis, peak strain and peak force can be found in Table 1, with the results of the ANOVA demonstrating that cholesterol accumulation may significantly affect $k_{50-100\%}$, AT hysteresis and AT force, but not AT strain.

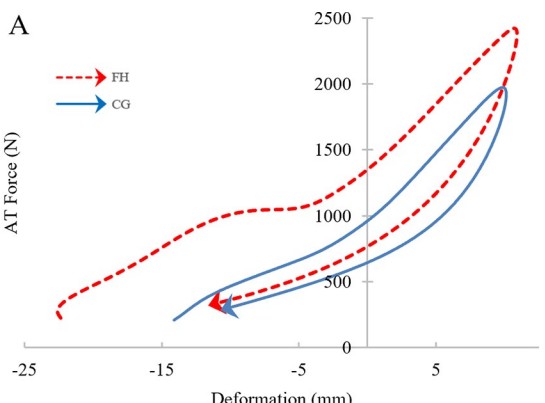
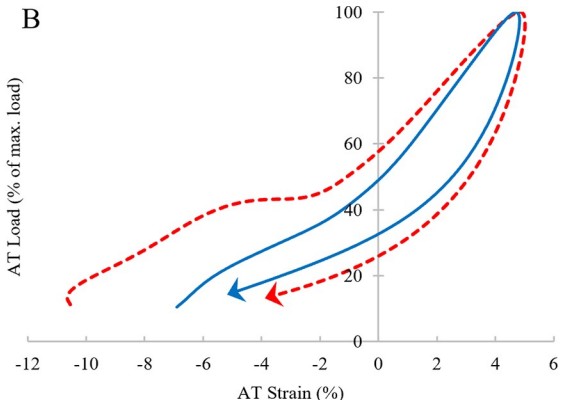

**Fig 1. Absolute force-deformation (A) and normalized force-strain curves (B).** (A) AT Force depicted in N and displacement in mm change in length relative to length at heelstrike. (B) Curve is normalized to allow for direct visual comparison, AT load is shown as percent of peak AT force and AT strain is shown as the percent change in length relative to length at heelstrike. Dashed line represents FH, solid line represents CG; arrow heads depict loop direction; data relating to AT force values below 200 N were removed for clarity (high GRF variability at low force levels).

SPM analysis revealed significantly higher AT forces ($p<0.001$) in FH during the early-mid phase (8–34%) of stance (Fig 2). Given this result, we calculated AT loading rates (using a 5% ground contact averaging window) and compared group values with an independent t-test and SPM analysis. The maximum AT loading rate occurred significantly earlier in the stance phase in FH than CG (14±5% vs. 20±6% of stance phase; $p = 0.0043$) and was significantly greater than in the CG (8.3±2.7 vs. 5.7±1.8 kN/sec; p = 0.003). SPM analysis of loading rate aligned with the t-test results, indicating an early difference between the two groups; SPM analysis also showed a significant difference between ~30–45% stance (Fig 3).

Whilst FH and CG report similar peak strains, the peak AT force is significantly greater in FH. It is subsequently surprising that FH stiffness ($k_{50-100\%}$) is lower than CG. This counterintuitive result requires a closer inspection of the AT force-length data, as the shape of the force curve—and hence load-strain curve–differed between groups (Fig 1B). $k_{cont}$ calculations were used to determine where stiffness differences occurred during the stance phase between groups, as this information is masked in the 50–100% AT force ranged used to calculate 'traditional' stiffness. CG had significantly increased $k_{cont}$ at 30% and 40% stance ($p = 0.001$ and $p = 0.002$, respectively), driven by the plateau in force measures in FH (Fig 4).

Females and males were equally affected by AT cholesterol accumulation in $k_{50-100\%}$ ($p = 0.886$, F = 0.021 and $\eta^2 = 0.001$), hysteresis ($p = 0.832$, F = 0.046, $\eta^2 = 0.002$), peak strain ($p = 0.683$, F = 0.17, $\eta^2 = 0.006$) and peak AT force ($p = 0.904$, F = 0.015, $\eta^2 = 0.001$). Sex was a not predictor for each of the main outcomes ($p>0.05$).

**Table 1. Descriptive statistics and significance levels, F statistic and partial eta-squared for AT variables.**

|  | FH Participants | CG Participants | *p*-value[a] | F | $\eta^2$ |
|---|---|---|---|---|---|
| Stiffness ($k_{50-100\%}$)* | 87.4±20.3 N/mm | 110.9±17.5 N/mm | 0.001 | 12.21 | 0.29 |
| Hysteresis* | 57.5±7.3% | 43.8±10.5% | <0.001 | 18.19 | 0.38 |
| Peak strain* | 5.26±1.2% | 4.95±0.9% | 0.413 | 0.69 | 0.02 |
| Peak AT force* | 2601±642 N | 2061±461 N | 0.01 | 7.46 | 0.2 |

* Mean ± SD.

[a]*p*-values obtained through one-way ANOVA.

**Abbreviations:** F = F-statistic; $\eta^2$ = partial eta-squared.

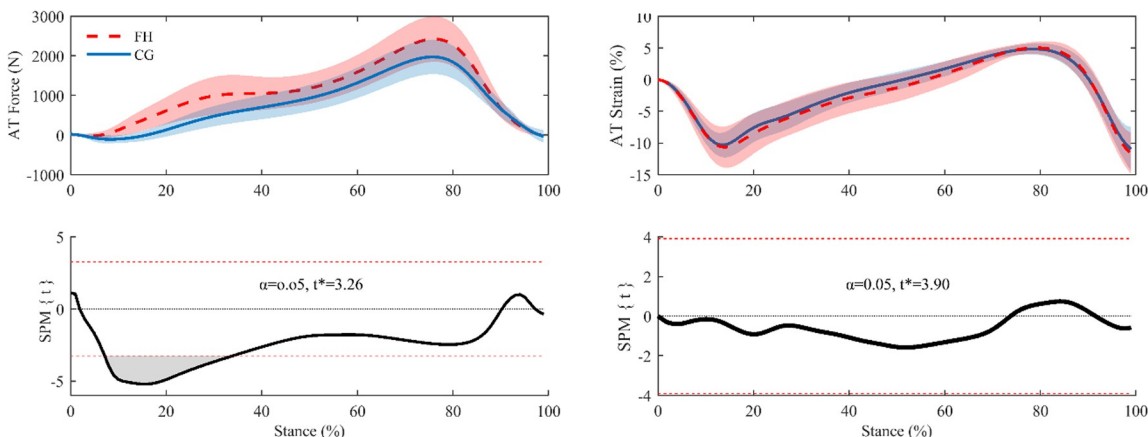

**Fig 2. AT force (left) and AT strain (right) mean and standard deviation (top), and corresponding SPM analysis (bottom).**
Bottom left displays a supra-threshold cluster (grey area) between 8% and 34% of the stance phase, indicating a significant difference in AT force between groups and the critical threshold ($t^* = 3.26$) as a red dashed line ($p<0.05$). Bottom right shows no difference in AT strain between groups throughout the stance phase.

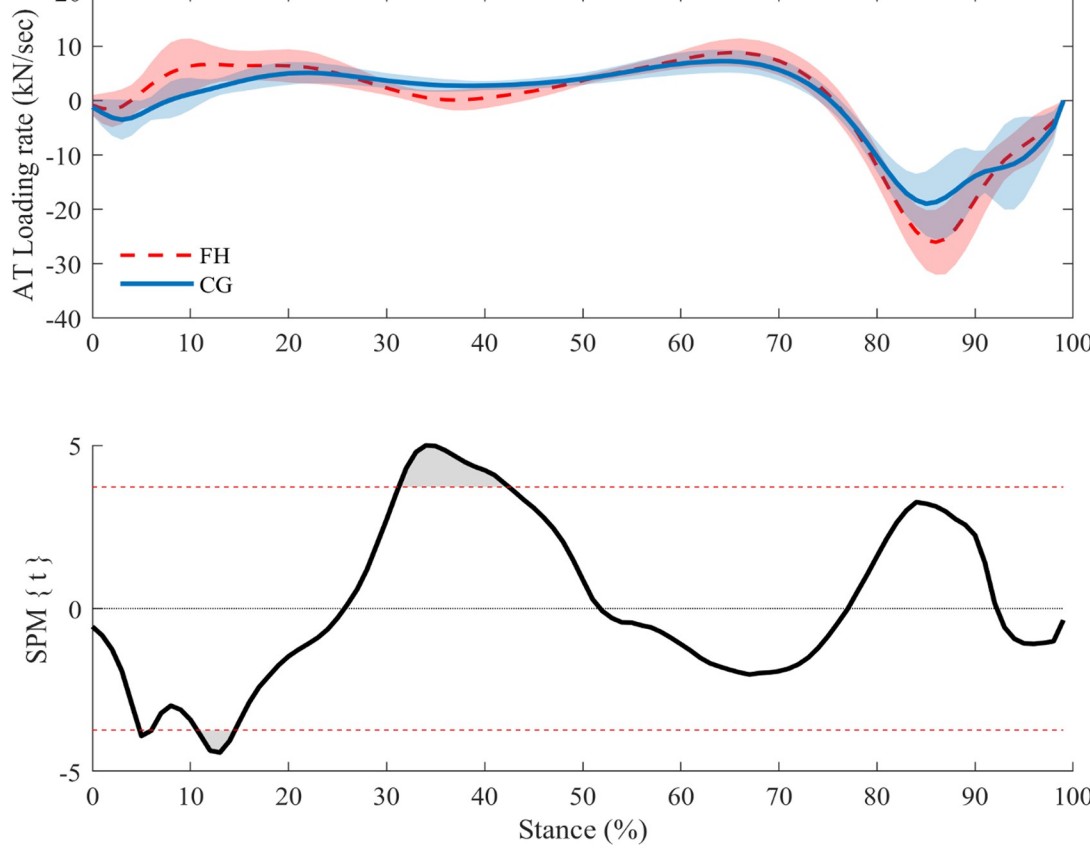

**Fig 3. AT loading rate mean and standard deviation (top) and SPM analyses (bottom).** $t^* = 3.72$ indicates a significant difference in loading rate at ~5%, ~10–15% and ~30–45% stance ($p<0.05$).

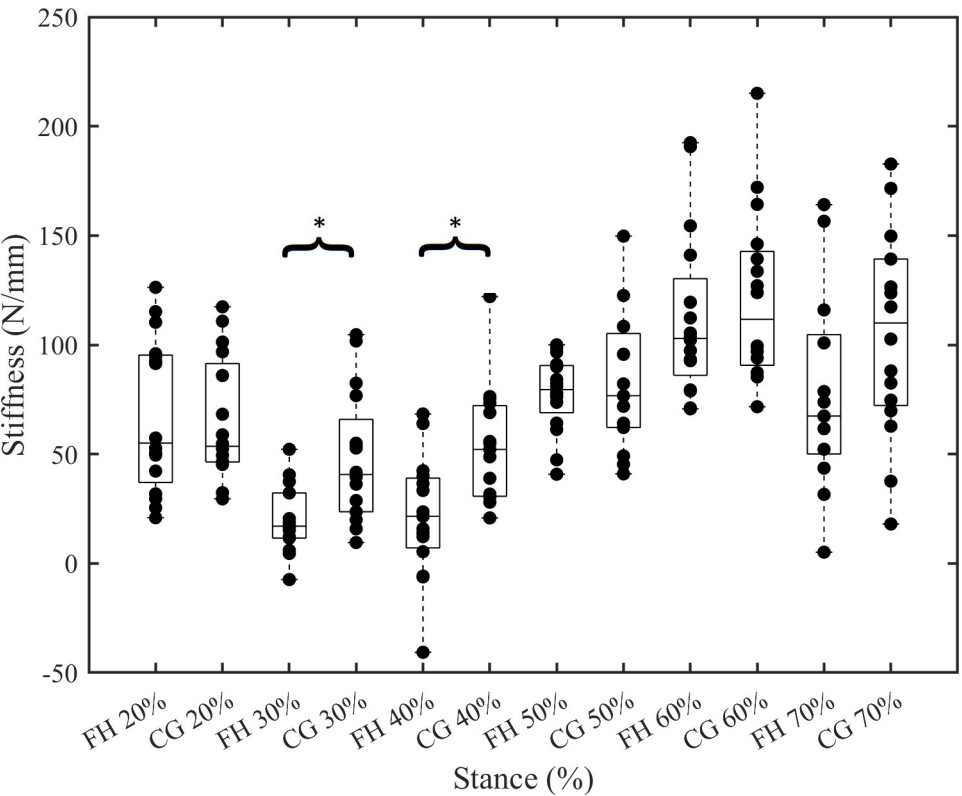

**Fig 4. Continuous stiffness from 20–70% stance.** $k_{cont}$ displayed in stance increments of 10%; arranged for inter-group comparison. $k_{cont}$ was significantly different at 30% and 40% stance; all others were not significant ($p = 0.065$–$0.301$). *indicates $p < 0.05$.

## Discussion

This study aimed to explore the impact of tendon cholesterol accumulation on AT biomechanical properties and behavior during walking using diagnosed TXT in persons with FH. Our FH group demonstrated significantly different AT biomechanics compared to CG during walking. FH exhibited a less linear ascending force-strain relationship, with a clear plateau in AT force from 30–40% stance, which drove significant differences in reported stiffness ($k_{50-100\%}$ and $k_{cont}$ at 30 and 40% stance). Hysteresis was also significantly increased in FH, confirming part of our hypothesis that stiffness and hysteresis would be altered in this group; however strain was not different between groups. These findings could be a result of cholesterol accumulation within the AT of these individuals. Previous data has shown that cholesterol accumulation results in less homogenous collagen fibre size and alignment, increased vascularity and increased LDL oxidation in tendon tissue leading to inflammation and failure at lower tendon forces [10, 13, 16, 31]. Such disruptions to AT composition could affect extension mechanisms and result in less efficient gait and increased injury risk.

In theory, should stiffness be reduced the tendon would deform more at a given force [32]. It is interesting that there was not a significant difference in strain between the two groups during the observed walking trials, given slightly greater AT forces in FH. It is possible that AT forces during walking were too low to reveal deformation differences that could be caused by cholesterol accumulation; peak forces are estimated to be between ~35–50% plantarflexion maximum voluntary contraction [33]. Perhaps in activities involving higher AT forces, such as hopping or maximum voluntary contractions, group differences in strain would be observed.

Nonetheless, a lower $k_{50\text{-}100\%}$ was seen in participants with FH, represented as the slope between 50–100% of the force-strain curve. This force range also aligned appropriately with the linear portion of the curve (Fig 1). Similar to our findings, tendinopathic tendons also display changes altered mechanics, such as decreased stiffness, increased strain and an increased cross-sectional area [32]. Arya and Kulig suggest that this altered morphology puts the AT at risk for additional injury and extended recovery time. Tendinopathic tendons have increased matrix separation, reduced expression and alignment of type I collagen, and increased type III collagen, all of which contribute to a fundamentally weaker tendon [34]. With a reduction in stiffness, the tendon is unable to resist higher forces which could result in experiencing higher strains and microtrauma [32]. Despite differences in morphology, FH AT show similar biomechanical features to tendinopathic tendon as a result of cholesterol accumulation. In conjunction with hypercholesterolemia research, our research provides a biomechanical basis that suggests that patients with FH could be at higher risk for tendinopathies, leading to higher frequencies of tendon pain compared to the general population [5–7, 35].

$k_{cont}$ was used to further describe AT stiffness in 10% stance phase increments between 20–70% of stance, and FH was found to have a significantly lower $k_{cont}$ at 30% and 40% stance (Fig 4). In order to investigate the underpinnings of these group differences, we performed SPM analysis of AT loading rate, which showed that there was a significant reduction in FH loading rate between 30–45% stance (this difference in loading rate is visible by way of a plateau in AT forces at mid-stance in FH; Fig 2A). As tendons are viscoelastic entities, their mechanical properties are rate-dependent. As such, we might expect tendon to be stiffer under higher-rate loading [36–38]. Consequently, the lower loading rates in FH tendons around mid-stance may help explain the lower $k_{cont}$ around the same time point. The AT force plateau may also help explain the difference in $k_{50\text{-}100\%}$ between groups, as it tended to occur around 50% of AT force and therefore may have been incorporated into the $k_{50\text{-}100\%}$ calculation and reduced the force-elongation slope.

Participants with FH also had a significantly higher maximum AT loading rate, with the maximum loading rate occurring earlier in the phase (10–15% stance). The altered loading pattern seems to cause the FH group to avoid normal unloading behaviors just after heel strike. In contrast, the CG AT predicably unloads when the tibialis anterior eccentrically contracts to control plantar flexion (Fig 2A, 0–20% stance) followed by continued loading throughout mid-stance [39]. To attempt to explain the altered AT loading phenomena, we worked backwards from AT force to identify which variable(s) used in its calculation may be responsible for the differences observed between groups. There were no notable differences in instantaneous AT moment arm length, but differences in plantarflexor moment led us to examine the variables used in calculating the external moment arm. As the ankle joint center is used in calculating AT moment arm length, which did not differ between groups, we concluded that differences must lie in the progression of the center of pressure. Interestingly, prior studies have shown that some conditions affecting the AT (e.g. diabetes, tendinopathy) can cause changes to center of pressure progression and therefore plantarflexion moment [23, 40–42]. Changes may be due to protective movement compensation cascading from tendon inefficiencies and discomfort, which is especially relevant since patients with FH suffer from increased rates of AT pain [35, 41]. Similarly, runners with AT tendinopathies may have increased maximum lateral and braking forces, as well as reduced time to maximum lateral and braking forces, at heel strike to fore foot contact [42]. Van Ginkle *et al.* suggest that these differences may be caused by the presence of a pronation heel-strike pattern that progresses to foot inversion, ultimately reducing the ability for the AT to absorb shock and exerting more stress through the tendon at an earlier time point [43]. Changes to tendinopathic plantar pressures are likely due to changes in onset, offset, duration and amplitude of muscle activity, however data are

inconsistent across most studies [41, 42]. Given the biomechanical similarities in the AT of people with tendinopathies and FH, it is feasible that plantar pressure changes may be at play in patients with FH, resulting in the altered loading phenomena. However, this level of analysis was beyond the scope of this study and should be investigated in future research studies.

Results of the current study show that FH had significantly higher AT hysteresis values than CG, which is visible in Fig 1, as the differences in the area between ascending and descending portions of the load-deformation curve. Interestingly, participants with FH returned around ~26% less stored AT energy than the CG participants. Reduced return of stored energy could lead to an increased metabolic cost of locomotion by demanding more work from muscles of the calf, knee, and hip [44]. The phenomena of increased hysteresis, or reduced return of stored energy, in participants with FH could be explained in several ways. First, AT TXT has been shown to have increased water content, which is associated with increased viscosity and therefore increases fluid efflux, removing more energy from the tendon in the form of heat [29, 45]. Second, poorly aligned collagen, cholesterol deposits, swelling, inflammation, and angiogenesis could affect the tendon's extension mechanisms, by increasing friction internally and externally, also increasing heat dissipation into surrounding tissues [8, 16, 22, 46]. Lastly, the musculotendinous unit could lose some of its ability to return elastic strain energy, owing to changes in AT stiffness [23, 47]. Historically, studies measuring AT hysteresis report values ranging from 2–49% depending on the activity and data collection methodology [27, 48, 49]. Hysteresis values recorded in higher loading rate activities, like one leg hopping (e.g., Lichtwark, 2005), tend to be smaller than lower loading rate activities like walking (e.g., Zelik & Franz, 2017) [27, 49]. Lower AT loading rates have been shown to enhance in vivo viscoelastic behavior, which may be a factor in the results of Zelik & Franz (2017), who demonstrated that slower walking velocity increased hysteresis values significantly from those observed with faster walking [37, 49]. As walking velocities were similar between groups in the current study, the chance that different velocities were responsible for the group differences are removed.

A few limitations are present in the study that could have influenced our results. First, patients with FH were diagnosed with TXT by a physician during a physical examination using binary grading (TXT or no TXT); therefore, the size of the TXT in our participants is unknown. Unfortunately, this negates our ability to report or comment on any correlations between severity of disorder or use of lipid lower medications and AT properties. Second, AT length was measured from the calcaneus marker (minus shoe thickness) to the instantaneous coordinates of the MTJ, which does not account for the anterior curve of the AT [50]. As AT TXT can cause the tendon to gain a convex shape in addition to increased anterior-posterior thickness, our groups may not be equally affected by this limitation [51]. However, as we did not calculate the thickness or cross-sectional area of the tendon (due to artificial diameter changes from cholesterol accumulation, stress could have been an unhelpful variant that may have detracted from our analysis), we were not able to account for the AT thickness. This feature could also have impacted the estimation of the AT moment arm in the FH group, which used a linear model of the AT force line of action. Lastly, sample size was relatively restricted due to the shortage of patients with AT TXT diagnosis within the BC FH Registry. The lack of TXT reporting within the registry is likely due prioritization of other clinical outcomes and under reporting of inconclusive physical examinations (potentially cases with diffuse thickening). Therefore, due to the small sample size, drawing conclusions based on the results of the study should be approached with caution.

Future research could be undertaken in the form of a larger longitudinal study, to determine if best treatment methods (lipid lowering medication, diet, and physical activity) can affect the negative impact of cholesterol accumulation on the structure and function of the AT, and to explore the potential relations among tendon function and severity of the disorder. In

addition, examining tendon function at higher loading rates or loads may reveal further effects of high cholesterol on relevant aspects of tendon function.

## Conclusion

This study investigated the impact of tendon cholesterol on Achilles tendon function during walking activity. The FH group displayed decreased AT stiffness compared to the control group, which appeared to be linked to AT loading rate rather than an increased strain. We found different AT force loading patterns in FH, which were traced to differences in the centre of pressure progression that affected ankle moment. This finding may indicate different AT loading strategies in FH to avoid pain. Participants with FH also demonstrated significantly greater AT hysteresis, suggesting that ambulation (or any activity involving a stretch-shortening cycle) may require a greater metabolic cost. It is possible that some of our results can be attributed to cholesterol accumulation within the tendon, altering its typical extension mechanisms. These findings represent a new contribution to our understanding of how cholesterol accumulation affects the function of the AT in people with FH. Our findings provide the basis for future in vivo studies exploring the biomechanical effects of cholesterol on tendons.

## Acknowledgments

We would like to thank The Centre for Hip Health and Mobility for providing the space, equipment, and technical assistance for conducting this project, Michael Zahradnik for assisting with data collection, Jiri Frohlich and Lubomira Cermakova for assisting with participant recruitment, and the BC FH Registry patients for their collaboration.

## Author Contributions

**Conceptualization:** Alexander Scott, Charlie M. Waugh.

**Data curation:** Kipling Squier, Charlie M. Waugh.

**Formal analysis:** Kipling Squier, Charlie M. Waugh.

**Funding acquisition:** Alexander Scott, Charlie M. Waugh.

**Investigation:** Kipling Squier.

**Methodology:** Kipling Squier, Michael A. Hunt, Charlie M. Waugh.

**Project administration:** Kipling Squier, Alexander Scott.

**Resources:** Kipling Squier, Alexander Scott, Liam R. Brunham, David R. Wilson, Charlie M. Waugh.

**Software:** Kipling Squier, Charlie M. Waugh.

**Supervision:** Alexander Scott, Charlie M. Waugh.

**Validation:** Michael A. Hunt, David R. Wilson, Charlie M. Waugh.

**Visualization:** Kipling Squier, Alexander Scott, Hazel Screen, Charlie M. Waugh.

**Writing – original draft:** Kipling Squier.

**Writing – review & editing:** Kipling Squier, Alexander Scott, Michael A. Hunt, Liam R. Brunham, David R. Wilson, Hazel Screen, Charlie M. Waugh.

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
