## [Decision Letter · Decision Letter 0]

13 Jul 2021

PONE-D-21-18535

The effects of cholesterol accumulation on Achilles tendon biomechanics: A cross-sectional study

PLOS ONE

Dear Dr. Squier,

Thank you for submitting your manuscript to PLOS ONE. After careful consideration, we feel that it has merit but does not fully meet PLOS ONE’s publication criteria as it currently stands. Therefore, we invite you to submit a revised version of the manuscript that addresses the points raised during the review process.

We look forward to receiving your revised manuscript.

Kind regards,

Tomoyoshi Komiyama, Ph.D

Academic Editor

PLOS ONE

Journal Requirements:

Additional Editor Comments:

Dear Authors,

The authors results show different Achilles loading patterns in the familial hypercholesterolemia group, which were traced to differences in the centre of pressure progression that affected ankle moment. This finding may indicate that individuals with familial hypercholesterolemia use different Achilles loading strategies. Participants with familial hypercholesterolemia also demonstrated significantly greater Achilles hysteresis than the control group, suggesting that walking may require a greater metabolic cost. Your results indicate that cholesterol accumulation could contribute to reduced Achilles function, while potentially increasing the chance of injury.

Authors should revise with reference to Reviewer's comments. In particular, the author needs to respond to reviwer3's comments. Originality is important, please check the link below.

I have carefully considered your manuscript up until now.

Unfortunately, I decided on major revision based on three reviewer’s comments.

It is unclear what this study reveals and what your results provide and what will be helpful for research in the future of medical research.

If these results become clear, I think your research is important for the future of clinicians and clinical researchers who aim to better understand your research.

Please check the PLOS ONE publishing standards.

https://journals.plos.org/plosone/s/criteria-for-publication

Criteria for Publication of the PLOS ONE

2. Results reported have not been published elsewhere.

Previously Published Studies

PLOS ONE does not accept for publication studies that have already been published, in whole or in part, elsewhere in the peer-reviewed literature. All figures included in manuscripts should be original, and should not have been published in any previous publications.

In addition, we will not consider submissions that are currently under consideration for publication elsewhere.

Reviewers' comments:

Reviewer's Responses to Questions

**Comments to the Author**

1. Is the manuscript technically sound, and do the data support the conclusions?

Reviewer #1: Yes

Reviewer #2: Yes

Reviewer #3: Partly

2. Has the statistical analysis been performed appropriately and rigorously? 

Reviewer #1: Yes

Reviewer #2: Yes

Reviewer #3: Yes

3. Have the authors made all data underlying the findings in their manuscript fully available?

Reviewer #1: Yes

Reviewer #2: Yes

Reviewer #3: No

4. Is the manuscript presented in an intelligible fashion and written in standard English?

Reviewer #1: Yes

Reviewer #2: Yes

Reviewer #3: Yes

5. Review Comments to the Author

Reviewer #1: This a very interesting study in an important issue. The results are clear and the discussion is according to the results. It is not clearly mentionned about cholesterol and apolipoprotein levels in the experimental but also in the control group. Moreover did the patients were under hypolipidemic treatment or did they received such a treatment in the past? This is an important issue as treatment may affect considerably tendons properties. A minor comment is the rather limited number of subjects in order to draw any definite conclusions and thus should be added in the limitations.

Reviewer #2: The authors of a manuscript entitled: “The effects of cholesterol accumulation on Achilles tendon biomechanics: A cross-sectional study” present comprehensive data on biomechanical properties of the Achilles tendons recorded during physical activities of a group of patients. This study provides a valuable contribution to understanding the impact of hypercholesterolemia on the structure and function of tendons in hypercholesterolemic patients. Unlike other studies on this subject that utilize suboptimal experimental systems, the authors utilize biologically and clinically relevant groups of patients, thereby increasing the overall value of this study.

Overall, the study is well designed and executed. The authors present the limitations of this study and justify them adequately. Addressing the following minor points would provide a piece of valuable information for a reader:

1. The authors should explain the impact of specific group sizes on observed differences.

2. The authors should comment if they observed any differences between the female and male participants.

3. Although the authors explained that they could not measure the extent of accumulation of lipids in the tendons, they should present data on serum concentration of the lipids if available.

Reviewer #3: Dear Authors,

In your study, you investigated the impact of tendon cholesterol on Achilles tendon function during walking activity. The FH group displayed decreased AT stiffness compared to the control group, which appeared to be linked to AT loading rate rather than an increased strain. Authors found different AT force loading patterns in FH, which were traced to differences in the centre of pressure progression that affected ankle moment.

These are my major comments for revision.

1. I read your previous manuscript published by the University of British Columbia.

“The effects of familial hypercholesterolemia on Achilles tendon biomechanics: a cross-sectional study (2018)”

I think it is necessary to clarify the relationship with this treatise in the introduction and state the differences from this previous treatise.

2. It seems that the methodology is different from the previous study.

Please explain, how and why particular analytic methods were selected.

3. Are there any alternative explanations for the results obtained by the analytical methods used?

4. Are there any unconventional approaches of data analysis used, and how do they differ from previous methods?

5. Did you analyze using the same patients as in the previous study?

Patient information should be stated as a supplement or specified in the method.

6. If the patient data is the same, additional samples need to be analyzed for the first time.

And will the results support the results of this study? If you do not need to add new any patients,

please explain the reasons.

6. PLOS authors have the option to publish the peer review history of their article (what does this mean?). If published, this will include your full peer review and any attached files.

Reviewer #1: No

Reviewer #2: No

Reviewer #3: No

---

## [Author Response · Author response to Decision Letter 0]

18 Aug 2021

Dear Dr. Komiyama and reviewers,

Thank you for your comments and questions regarding “The effects of cholesterol accumulation on Achilles tendon biomechanics: A cross-sectional study”. I thank you for your encouraging words surrounding the importance/interesting nature of this exploratory study.

Reviewer #1.

“It is not clearly mentioned about cholesterol and apolipoprotein levels in the experimental but also in the control group.”

I have collected clinical characteristics for the participants with FH from their yearly visit to the FH clinic (results closest to their data collection date), CG clinical characteristics were not able to be collected. However, each CG participant stated that they were healthy (to the best of their knowledge) and were not currently taking any form of medication that may confound our findings. 

Added revision at line #225. “Clinical characteristics for participant’s with FH (CG not available) retrieval revealed a mean Dutch Lipid Clinic Network Score of 18±6 and a moderately controlled lipid profile (LDL: 3.5±1.6 mmol/L, total cholesterol: 5.4±1.6 mmol/L, apolipoprotein B:1.02±0.36 g/L).” 

“Did the patients were under hypolipidemic treatment or did they received such a treatment in the past? This is an important issue as treatment may affect considerably tendons properties.”

All but one of the participants with FH were currently using lipid lowering medications to manage the disorder. This was expected as the more serious FH cases tend to present with xanthoma (adds an extra 6 points on to the Dutch lipid clinic network score, which explains why participants with FH had a mean score of 18). In the limitations, I have stated that it is not possible for this study to report correlations between severity of disorder or use of lipid lower and AT properties, due to the unknown size or severity of the AT TXT. 

I have added a revision at line #222. “Each participant with FH was noted to be positive for TXT within the registry medical charts and all but one participant with FH were currently using one or more lipid lowering medications (mean usage 7.7±7.2 years).”

“A minor comment is the rather limited number of subjects in order to draw any definite conclusions and thus should be added in the limitations.”

Added a revision at line #394. “Lastly, sample size was relatively restricted due to the shortage of patients with AT TXT diagnosis within the BC FH Registry. The lack of TXT reporting within the registry is likely due prioritization of other clinical outcomes and under reporting of inconclusive physical examinations (potentially cases with diffuse thickening). Therefore, due to the small sample size, drawing conclusions based on the results of the study should be approached with caution.

Reviewer #2.

1. “The authors should explain the impact of specific group sizes on observed differences.”

Added revision at line #394. “Lastly, sample size was relatively restricted due to the shortage of patients with AT TXT diagnosis within the BC FH Registry. The lack of TXT reporting within the registry is likely due prioritization of other clinical outcomes and under reporting of inconclusive physical examinations (potentially cases with diffuse thickening). Therefore, due to the small sample size, drawing conclusions based on the results of the study should be approached with caution.”

2. “The authors should comment if they observed any differences between the female and male participants.”

Added revision at line #217. “MANOVA analysis was performed to determine if sex played a role in the effect of cholesterol accumulation on the primary outcomes”.

Added revision at line #284. “Females and males were equally affected by AT cholesterol accumulation in k50-100% (p=0.886, F=0.021 and η2=0.001), hysteresis (p=0.832, F=0.046, η2=0.002), peak strain (p=0.683, F=0.17, η2=0.006) and peak AT force (p=0.904, F=0.015, η2=0.001). Sex was a not predictor for each of the main outcomes (p>0.05).”

3. “Although the authors explained that they could not measure the extent of accumulation of lipids in the tendons, they should present data on serum concentration of the lipids if available.”

Added revision at line #225. “Clinical characteristics for participant’s with FH (CG not available) retrieval revealed a mean Dutch Lipid Clinic Network Score of 18±6 and a moderately controlled lipid profile (LDL: 3.5±1.6 mmol/L, total cholesterol: 5.4±1.6 mmol/L, apolipoprotein B:1.02±0.36 g/L).” 

Reviewer #3.

Thank you for your comments, the manuscript (The effects of familial hypercholesterolemia on Achilles tendon biomechanics: a cross-sectional study [2018]) held within University of British Columbia’s online repository (cIRcle) is an unpublished master’s thesis. The purpose of cIRcle is to act an open access digital repository for unpublished material produced by students at the university. Thesis submission to cIRcle is a graduation requirement for every thesis-based student at the university. 

Alterations were made to the current submission (compared to the thesis) to a) address the results/discussion in a more concise and appropriate way, making it suitable for peer reviewed publication, b) consider concerns/suggestions from co-authors that had not been a part of the master’s committee. 

I look forward to hearing your feedback concerning the revisions and thank you for your time.

Kip Squier

---

## [Decision Letter · Decision Letter 1]

31 Aug 2021

The effects of cholesterol accumulation on Achilles tendon biomechanics: A cross-sectional study

PONE-D-21-18535R1

Dear Dr. Squier,

We’re pleased to inform you that your manuscript has been judged scientifically suitable for publication and will be formally accepted for publication once it meets all outstanding technical requirements.

Kind regards,

Tomoyoshi Komiyama, Ph.D

Academic Editor

PLOS ONE

Additional Editor Comments (optional):

Dear author,

Thank you for submitting your revised manuscript.

I think it was much easier to understand than the original manuscript.

I am satisfied with the responses and the edits, I am happy to accept this manuscript.

The authors have replied to my remaining comments satisfactorily from two reviewers.

Therefore, I have no further comments to make, all of my previous concerns were adequately addressed.

This manuscript will be satiating the reader's interest.

Best regards,

Tomoyoshi Komiyama

Reviewers' comments:

Reviewer's Responses to Questions

**Comments to the Author**

1. If the authors have adequately addressed your comments raised in a previous round of review and you feel that this manuscript is now acceptable for publication, you may indicate that here to bypass the “Comments to the Author” section, enter your conflict of interest statement in the “Confidential to Editor” section, and submit your "Accept" recommendation.

Reviewer #2: All comments have been addressed

Reviewer #3: All comments have been addressed

2. Is the manuscript technically sound, and do the data support the conclusions?

Reviewer #2: Yes

Reviewer #3: Yes

3. Has the statistical analysis been performed appropriately and rigorously? 

Reviewer #2: Yes

Reviewer #3: Yes

4. Have the authors made all data underlying the findings in their manuscript fully available?

Reviewer #2: Yes

Reviewer #3: Yes

5. Is the manuscript presented in an intelligible fashion and written in standard English?

Reviewer #2: Yes

Reviewer #3: Yes

6. Review Comments to the Author

Reviewer #2: (No Response)

Reviewer #3: Dear author,

Thank you for submitting your reply comments and revised manuscript.

I have no further questions and I understand the previous history this manuscript.

7. PLOS authors have the option to publish the peer review history of their article (what does this mean?). If published, this will include your full peer review and any attached files.

Reviewer #2: No

Reviewer #3: No

---

## [Editor Report · Acceptance letter]

8 Sep 2021

PONE-D-21-18535R1 

The effects of cholesterol accumulation on Achilles tendon biomechanics: A cross-sectional study 

Dear Dr. Squier:

I'm pleased to inform you that your manuscript has been deemed suitable for publication in PLOS ONE. Congratulations! Your manuscript is now with our production department. 

Kind regards, 

on behalf of

Dr. Tomoyoshi Komiyama 

Academic Editor

PLOS ONE